# Effect of the Ammonium Tungsten Precursor Solution with the Modification of Glycerol on Wide Band Gap WO_3_ Thin Film and Its Electrochromic Properties

**DOI:** 10.3390/mi11030311

**Published:** 2020-03-16

**Authors:** Jinxiang Liu, Guanguang Zhang, Kaiyue Guo, Dong Guo, Muyang Shi, Honglong Ning, Tian Qiu, Junlong Chen, Xiao Fu, Rihui Yao, Junbiao Peng

**Affiliations:** 1Institute of Polymer Optoelectronic Materials and Devices, State Key Laboratory of Luminescent Materials and Devices, South China University of Technology, Guangzhou 510640, China; 201765340428@mail.scut.edu.cn (J.L.); msgg-zhang@mail.scut.edu.cn (G.Z.); 201836320090@mail.scut.edu.cn (K.G.); 201430320229@mail.scut.edu.cn (M.S.); msjlchen@gmail.com (J.C.); 201630343721@mail.scut.edu.cn (X.F.); psjbpeng@scut.edu.cn (J.P.); 2School of Medical Instrument & Food Engineering, University of Shanghai for Science and Technology, No.516 Jungong Road, Shanghai 200093, China; guodong99@tsinghua.org.cn; 3Department of Intelligent Manufacturing, Wuyi University, Jiangmen 529000, China; timeqiu@hotmail.com; 4Guangdong Province Key Lab of Display Material and Technoloy, Sun Yat-sen University, Guangzhou 510275, China

**Keywords:** tungsten trioxide film, spin coating, optical band gap, morphology, electrochromism

## Abstract

Tungsten trioxide (WO_3_) is a wide band gap semiconductor material, which is commonly not only used, but also investigated as a significant electrochromic layer in electrochromic devices. WO_3_ films have been prepared by inorganic and sol-gel free ammonium tungstate ((NH_4_)_2_WO_4_), with the modification of glycerol using the spin coating technique. The surface tension, the contact angle and the dynamic viscosity of the precursor solutions demonstrated that the sample solution with a 25% volume fraction of glycerol was optimal, which was equipped to facilitate the growth of WO_3_ films. The thermal gravimetric and differential scanning calorimetry (TG-DSC) analysis represented that the optimal sample solution transformed into the WO_3_ range from 220 °C to 300 °C, and the transformation of the phase structure of WO_3_ was taken above 300 °C. Fourier transform infrared spectroscopy (FT-IR) spectra analysis indicated that the composition within the film was WO_3_ above the 300 °C annealing temperature, and the component content of WO_3_ was increased with the increase in the annealing temperature. The X-ray diffraction (XRD) pattern revealed that WO_3_ films were available for the formation of the cubic and monoclinic crystal structure at 400 °C, and were preferential for growing monoclinic WO_3_ when annealed at 500 °C. Atomic force microscope (AFM) images showed that WO_3_ films prepared using ammonium tungstate with modification of the glycerol possessed less rough surface roughness in comparison with the sol-gel-prepared films. An ultraviolet spectrophotometer (UV) demonstrated that the sample solution which had been annealed at 400 °C obtained a high electrochromic modulation ability roughly 40% at 700 nm wavelength, as well as the optical band gap (E_g_) of the WO_3_ films ranged from 3.48 eV to 3.37 eV with the annealing temperature increasing.

## 1. Introduction

Currently, an increasing amount of attention has been concentrated on energy saving in buildings for the reason that the energy used for these is in excess of 30% of the consumption of the total energy in the word, as a matter of fact [1]. It is worth mentioning that it is the transition metal oxides that are the fascinating semiconducting materials which possess a variety of properties, applications and functions [2]. Tungsten trioxide (WO_3_), the most widely studied and used electrochromic material, has been studied for many years since having been found by Deb in the 1960s, thanks to its attractive characteristics of high coloring efficiency, good optical modulation ability and decent chemical stability in the field of electrochromism [3,4,5,6]. It is remarkable that WO_3_ with various crystalline structures by different synthesis methods is extensively appropriate for diverse device applications [7,8,9], such as the electrochromic smart window [10,11], rear-view automatic-dimming rearview mirror [12], gas sensor device [13,14,15], and military camouflage [16], etc. A large variety of techniques can be used to prepare WO_3_ thin films, such as chemical vapor deposition (CVD) [17], physical vapor deposition (PVD) [18] and wet chemical deposition methods base on solution [19,20,21]. In fact, the sol-gel technique that is used to prepare WO_3_ thin films is commonly considered feasible among wet chemical deposition methods due to a good few favorable and conspicuous advantages, such as large-area film formation, a repeatable process and inexpensive experimental facilities [22,23,24]. It is the formation of the gel network by polymerization that the principle of the sol-gel technique for preparing thin films depends on [3]. Nevertheless, it cannot be neglected that there is less satisfaction with thin films prepared by the sol-gel technique because of these following deficiencies, which would be limited in applications in electrochromism and thin film transistors, etc. Not only the stability of the precursor solution for the formation of gel structure is generally not adequate enough, and it would not gratify the industrial production as a significant parameter [25], but also the homogeneity throughout the sol-gel process would not be guaranteed effectively by producing a homogeneous precursor solution at room temperature [26]. Furthermore, there are practical limitations for the reason that thin films with microstructures and nanostructures prepared by the sol-gel technique are more fragile, so that they possibly cannot maintain well their structure in the course of the assembly process of electrochromic devices [27]. It is important to note that wrinkle cracks are observed on the surface of the thin films prepared by the sol-gel technique with the increase in the thickness of films [28,29].

To effectively solve this problem, WO_3_ thin films can be prepared by the spin coating technique using pure-inorganic ammonium tungstate precursor solution in this present study, which would not obtain the gel structure. In this work, WO_3_ thin films were successfully prepared using the above precursor solution by the spin coating technique. The glycerol, one kind of excellent solvent with high viscosity, has been utilized to magnify the adhesiveness of the pure-inorganic ammonium tungstate precursor solution, so as to sufficiently facilitate the growth of the WO_3_ films. The sample solution performance, the film surface morphology, components, crystallization, optical properties and the electrochromic properties, were investigated and discussed through different characterization methods.

## 2. Materials and Methods 

Tungsten trioxide (WO_3_, <100 nm, 99.9% metals basis, Macklin Biochemical Co. Ltd, Shanghai, China) and ammonia hydroxide (NH_3_·H_2_O, AR, 26%, Guangzhou Chemical Reagent Factory, Guangzhou, China) were mixed in a beaker to prepare an ammonium tungstate ((NH_4_)_2_WO_4_) solution. Subsequently, five kinds of sample solution were prepared using glycerol (C_3_H_8_O_3_, AR, Richjoint, Shanghai, China) for modification. According to the different volume fraction of glycerol, the volume fractions of glycerol were 0%, 12.5%, 25%, 37.5% and 50%, respectively, and the concentration of ammonium tungstate solution was approximately 5.5 mol/L. The precursor solutions were uniformly obtained after being ultrasonically oscillated, which were not only transparent, but also sol-gel free. Eventually, all of the WO_3_ thin films were prepared, using the spin coating technique, onto indium tin oxide (ITO) plane glass substrates (2 × 2 cm^2^), with spin coating parameters (3500 revolutions per minute for 60 s). 

The WO_3_ thin films were annealed at different temperatures in the air atmosphere: 200 °C, 250 °C, 300 °C, 350 °C, 400 °C, 450 °C and 500 °C, respectively.

The surface tension and the contact angle of the precursor solutions were measured by Attension Theta (Biolin Scientific, TL200, Gothenburg, Sweden). The dynamic viscosity measurements of the precursor solutions were implemented using a rotational rheometer (Thermo Fisher Scientific, HAAKE MARS 40, MA, USA). Thermal gravimetric analysis (TGA) and differential scanning calorimetry were characterized by a Differential Scanning Calorimeter (DSC, Differential Scanning Calorimeter, DSC214, NETZSCH Scientific Instruments Trading (Shanghai) Ltd., Selb, Germany). The thickness of all films was measured by a probe surface profiler (VeecoDektak150, Veeco, Somerset, NJ, USA). The crystalline structure of WO_3_ thin films was characterized by X-ray Diffraction using Cu Kα radiation (XRD, PANalytical Empyrean DY1577, PANalytical, Almelo, The Netherlands), and the XRD patterns were analyzed using the software Jade 6.0. As well as this, Fourier transform infrared spectroscopy (FT-IR) spectra of the solutions were recorded by an FT-IR spectrophotometer (SHIMADZU IR Prestige-21, SHIMADZU, Tokyo, Japan) with an ITO glass substrate acting as a blank. The surface morphology was observed by an atomic force microscopy (AFM, Being Nano-Instruments BY3000, Being Nano-Instruments, Beijing, China). The transmittance of the films at the initial state, colored state and bleached state were measured by an ultraviolet spectrophotometer (SHIMADZU UV2600, SHIMADZU, Tokyo, Japan), with air acting as a blank. The current of the electrochromic test and the relationship between the change of transmittance and the time were recorded by an electrochemical workstation (CH Instruments CHI600E, CH Instruments, Shanghai, China) and a micro-spectrometer (Morpho PG2000, Morpho, Shanghai, China), respectively.

## 3. Results and Discussion

The surface tension, the contact angle and the dynamic viscosity of all the precursor solutions are illustrated in Figure 1. Figure 1a shows that as the volume fraction of modified-ammonia tungstate glycerol increases, keeping the volume of the precursor solutions consistent, both of the surface tension and the contact angle decrease first, and then increase while obtaining a minimum in the sample solution with 25% volume fraction glycerol. As shown in Figure 1b, the dynamic viscosity of the solutions characterizes an increasing tendency with the increase in the volume of the glycerol. As defined, when the droplet has a tendency to spread out on the solid surface, not only the solid–liquid contact surface increases and the contact angle decreases, but also the surface tension decreases. In other words, the smaller the surface tension and the contact angle, the better the surface wettability of the solution on the substrate. Additionally, bigger dynamic viscosity could facilitate the growth of films. In comparison to dynamic viscosity, the surface tension and the contact angle exert a decisive part in selecting the optimal solution in this work. As a consequence, the ammonium tungstate solution with 25% volume fraction glycerol is used to grow WO_3_ thin films in this work.

Figure 2 shows the thermal gravimetric and differential scanning calorimetry (TG-DSC) curves of the sample solution with 25% volume fraction glycerol and the thickness of the films annealed at different temperatures. The sample solution successively loses weight from room temperature to approximately 300 °C, as shown in the TG curve. The drastic weight loss can be as a result of the water evaporation range from room temperature to 140 °C [30]. It was the pyrolysis of glycerol and the further evaporation of water that a continuous weight loss of the solution range from 140 °C to 220 °C results from [31]. It can be seen that there are multiple weak endothermic peaks and exothermic peaks, and the weight of the sample solution reduces in the temperature between 220 °C and 300 °C, which is presumably ascribed to the formation of WO_3_ from ammonium tungstate. The weight of the sample solution was not observed to change obviously above 300 °C. Besides, the thickness of the WO_3_ films decreases drastically between 200 °C and 300 °C, as shown in Figure 2b. The drastic drop of thickness probably results from the water evaporation and the pyrolysis of glycerol. The further change in thickness would be analyzed through the FT-IR and XRD. Furthermore, the slowly endothermic process between 300 °C and 460 °C could be observed, which is possibly attributed to the transformation of phase structure of WO_3_ from ammonium tungstate. 

On the other hand, according to DSC analysis, it is concluded that the phase structure of WO_3_ was also transformed along with exothermic process above 460 °C. The further phase structure transformation of WO_3_ would be characterized and analyzed through XRD measurement.

The FT-IR spectra of the sample solution with a 25% volume fraction of glycerol are illustrated in Figure 3, which were annealed at 300 °C, 350 °C, 400 °C, 450 °C and 500 °C, respectively. It can be seen that there are a sharp peak at approximately 980 cm^−1^ and a weak peak at around 690 cm^−1^, which respectively correspond to the characteristic W=O stretching vibration and W–O–W stretching vibration [32,33], revealing the successful formation of WO_3_ from ammonium tungstate when WO_3_ films were annealed above 300 °C, which is consistent with the result of the TG-DSC analysis. In addition, the sharp peak shifts to a higher wavenumber, and becomes sharper from 300 °C to 350 °C in indication of that the W=O bond length shortens and the W=O bond vibrational steric resistance increases. It is indicated that the film structure becomes dense from loose, and the thickness of film deceases at 350 °C. Besides, the decrease in the film thickness in the temperature stage of 400–500 °C is also attributed to the densification of the film in the case of the stable content of WO_3_ in the film. As shown in Figure 3, both of the two peaks were intensified with the increase in the annealing temperature, which indicated that a mounting number of WO_3_ was generated, and the WO_3_ thin film was prone to be purer. According to FT-IR analysis, WO_3_ is the dominant composition within the films instead of glycerol as the annealing temperature increases.

Figure 4a–f presents the AFM 2D and 3D images (5000 nm × 5000 nm) of the WO_3_ thin films prepared using the sample solution with 25% volume fraction glycerol, which were annealed at different temperatures for an hour. Moreover, the surface roughness and the surface grain size of these films are illustrated in Figure 4g. As shown in Figure 4a,b, it is revealed that the WO_3_ thin film which was annealed at 200 °C is almost smooth and homogeneous. There are few surface grains on the WO_3_ thin films annealed at 200 °C for the reason that definite boundary would not be observed. According to the result of TG-DSC analysis, the film annealed at 200 °C did not transform into WO_3_ yet. With the increase in the annealing temperature, a slightly rough surface is commenced to emerge on the WO_3_ thin film, which can be observed in Figure 4c,d. There are a great many easily identifiable grains on the surface of the WO_3_ film in Figure 4e,f, which may be attributed to the crystallization of WO_3_ when the WO_3_ film was annealed at 500 °C. This can be confirmed through the result of XRD analysis. The average surface grain size of the WO_3_ thin films annealed at 200 °C, 350 °C and 500 °C are, respectively, 7.8 nm, 63.6 nm and 74.7 nm. The surface root mean square (RMS) roughness of the WO_3_ films annealed at 200 °C, 350 °C and 500 °C are 0.38 nm, 1.01 nm and 1.68 nm, respectively. It is worth mentioning that the WO_3_ films prepared in this work are less rough, with a surface roughness of less than 2 nm in comparison with the sol-gel-prepared WO_3_ films, although the surface roughness of WO_3_ films exists a slowly increasing tendency with annealing temperature [34,35,36]. In other words, the WO_3_ films with less roughness prepared in this work can reduce scattering of the incident light, and are beneficial to the transmission of incident light, so that they are more qualified for application in an electrochromic device [34].

X-ray Diffraction (XRD) is an effective experimental technique to characterize the crystalline structure and the grain size of material. The XRD patterns of the WO_3_ films prepared using the sample solution with 25% volume fraction glycerol onto ITO glass substrate, and annealed at different temperatures, are illustrated in Figure 5. The diffraction peaks were analyzed using Jade 6.0 and PDF#06-0416, PDF#41-0905 and PDF#30-1387. In the XRD of Figure 5, the diffraction peaks of the WO_3_ thin films annealed at 200 °C and 300 °C are matched well with Indium Oxide (In_2_O_3_), according to PDF#06-0416, indicating acquirement of the amorphous structure of the WO_3_ film [20,25,37]. It can be confirmed that there are some diffraction peaks of the WO_3_ film annealed at 400 °C, except for the diffraction peaks of ITO glass substrate in Figure 5. In other words, it is indicated that the crystalline temperature for the WO_3_ thin films is between 300 °C and 400 °C [2,38]. According to the broad diffraction peak located at approximately 24°, the formation of the WO_3_ cubic crystal structure is revealed, and monoclinic WO_3_ also exists at the same time [32]. It is suggested that the crystalline transformation in thin film, making the uplift of the film, results in the increase in thickness at 400 °C. Along with temperature being further increased, the WO_3_ film was confirmed to be preferential growth to monoclinic crystalline structure, rather than cubic crystalline structure when the WO_3_ film was annealed at 500 °C [16,39]. Furthermore, this can be in good agreement with the results of TG-DSC curves and AFM images.

Figure 6a–g display the optical transmittance spectra of the WO_3_ films at the initial state, colored state and bleached state in the wavelength range from 400 nm to 800 nm, which were prepared using the sample solution with 25% volume fraction glycerol, and annealed at different temperatures, and the optical transmittance modulation ability (∆T at 700 nm) curve of these films is illustrated in Figure 6h. The transmittance modulation ability (∆T) can be defined by the following formula at 700 nm wavelength:∆T=|T_c_ − T_b_|(1)

In this formula, T_c_ and T_b_ are the optical transmittance of the WO_3_ films at a colored state and bleached state at the 700 nm wavelength, respectively. It is meant that the greater the ∆T, the better the optical transmittance modulation ability. As shown in Figure 6h, ∆T increased first from 200 °C to 300 °C, and subsequently decreased slightly at 350 °C. After ∆T achieved a maximum roughly 40% at 400 °C, it decreased again at 450 °C and 500 °C.

In the temperature stage of 200–300 °C, the precursor has not completely transformed. The quite thick film is attributed to a large quantity of amorphous carbon from the pyrolysis of glycerol. Besides, the amorphous carbon is the good binding receptor for Li^+^ [40,41], thereby there is a competitive relationship between amorphous carbon and WO_3_ in binding to Li^+^. As the annealing temperature increases, the thickness of the film decreases, suggesting that the amorphous carbon is gradually oxidized, and the content of amorphous carbon decreases. WO_3_ gradually plays a dominant role in the competition of binding to Li^+^ with the continuous formation of tungsten trioxide. As a consequence, the ∆T of the film gradually increases range from 200 °C to 300 °C. 

When the annealing temperature rises above 350 °C, the composition of the film is basically WO_3_, and the effect of temperature on the mass of WO_3_ is negligible, according to the result of the TG curve in Figure 2. The ∆T of the film decreases slightly at 350 °C, which is attributed to the following possible reason: In the FT-IR of Figure 3, when the annealing temperature increases from 300 °C to 350 °C, the W=O absorption band shifts to a higher wavenumber and becomes sharper, which indicates that the W=O bond length shortens, as well as the W=O bond vibrational steric resistance increases. In addition, the mass of WO_3_ is basically constant, while the thickness of film decreases. It demonstrates that the film structure becomes dense from loose. The steric hindrance of Li^+^ into and out the thin film increases, and the binding sites may decrease at the same time, which is not conducive to electrochromism [42,43,44]. Therefore, there is a slight decrease in the ∆T at 350 °C. 

The ∆T of the film at 400 °C is the best for the possible reasons as follows: Firstly, the crystalline WO_3_ film possesses a higher carrier mobility [37,45], WO_3_ is an n-type semiconductor [2] and electrons are the majority carrier, which is beneficial to electron injection in the electrochromic double injection of ions and electrons model [28,46]. Secondly, the crystallization degree of the film is not very high at 400 °C, and there are a lot of grain boundaries within the film in a mixed crystalline phase structure. Besides, different crystalline phases would produce a lot of crystallographic defects owing to the lattice constant mismatch. These grain boundaries and defects are essentially W–O-dangling bond, which are good ion binding sites, contributing to electrochromism. Hence, the ∆T of the film is the best at 400 °C.

However, with the further increase in annealing temperature, the crystal phase of the film is mainly monoclinic, and the diffraction peaks become sharper in Figure 5, which indicates that the crystallization degree of the film increases, suggesting that the grain boundaries in the film decrease and Li^+^ binding sites decrease. Moreover, the densification of the film is not conducive to the injection and extraction of Lithium ions [44,47]. In consequence, the ∆T of thin film decreases gradually in the temperature stage of 400–500 °C. 

The optical properties of the WO_3_ films annealed at different temperatures are characterized in Figure 7. The optical band gap of these films can be calculated from the following formula [48,49]:αhν = A(hν − E_g_)^n^(2)

In this formula, A is a content; α, the absorption coefficient, can be computed from the transmittance; the Planck constant (h) is 6.626 × 10^−34^ J s; ν, the photon frequency, can be converted from the wavelength; n is 2, because WO_3_ is an indirect band gap semiconductor; E_g_, the optical band gap, is the value of the fitting line horizontal intercept of the curve of (αhν)^1/2^ versus the photon energy hν.

As shown in Figure 7b, the optical band gap (E_g_) of the WO_3_ films are observed at the 3.48 eV, 3.45 eV, 3.47 eV and 3.37 eV ranges from 350 °C to 500 °C, respectively. The E_g_ of the WO_3_ films prepared in this work decreased first and then increased, and afterward decreased again with the increase in annealing temperature. The average bandgap of the WO_3_ films with a mixed crystalline structure, obtaining a minimum, is 3.37 eV at 500 °C [2].

Figure 8a,b illustrate the change of transmittance at 680 nm wavelength of the WO_3_ films prepared using the sample solution with 25% volume fraction glycerol and the change of transmittance at 680 nm wavelength under ±3.0 V voltages, respectively. Additionally, for sufficiently evaluating the electrochromic performance, coloration efficiency (CE) as an important parameter is defined by the following formula [50,51]:CE = ΔOD/ΔQ = log[T_b_/T_c_]/(Q/A)(3)
Q = ∫ I dt(4)

In the formula, ΔOD is the optical density of the change between two optical states at 680 nm wavelength; T_b_ and T_c_ were defined above; the injecting and extracting charge density (Q) were calculated by integrating the current of a cycle of the electrochromic test; A is the electrochromic total area. The coloration efficiency (CE) value of sample was calculated to be 46.3 cm^2^·C^−1^ for WO_3_ thin films annealed at 400 °C in this work, as shown in Figure 8b. In general the CE value of sol-gel-prepared films ranges from 25 cm^2^·C^−1^ to 75 cm^2^·C^−1^, which shows that the CE value of sol-gel-free WO_3_ films as reliable as normal sol-gel WO_3_ films [37,52]. Further studies will be done about the electrical resistance and the electrochromic reliability.

It is worth comparing the performance in this work with other reported studies by other synthesis methods. The comparison of coloration efficiency and optical band gap between this work and other reported works would be demonstrated in Table 1. It is indicated that this work, using the sol-gel-free method, maintains a proper balance between optical band gap and coloration efficiency better than other works by synthesis methods in reference.

## 4. Conclusions

The WO_3_ thin films were successfully prepared on ITO glass substrate by spin coating technique using the glycerol-modified ammonium tungstate precursor solution. For better surface wetting, the sol-gel free ammonium tungstate solution with the 25% volume fraction of glycerol was selected to promote the growth of WO_3_ thin films. It is concluded that the ammonium tungstate transformed to WO_3_ range from 220 °C to 300 °C, and the crystalline structure of the WO_3_ films was observed when films were annealed above 300 °C.

Additionally, as the annealing temperature increases, the component content of WO_3_ within the film increased. The WO_3_ films prepared in this work were smoother and more homogeneous than the sol-gel-prepared films. The optical band gap of the WO_3_ films was variational from 3.48 eV to 3.37 eV with the increase in the annealing temperature above 350 °C. Furthermore, the coloration efficiency of the WO_3_ films attained 46.3 cm^2^·C^−1^.

## Figures and Tables

**Figure 1 micromachines-11-00311-f001:**
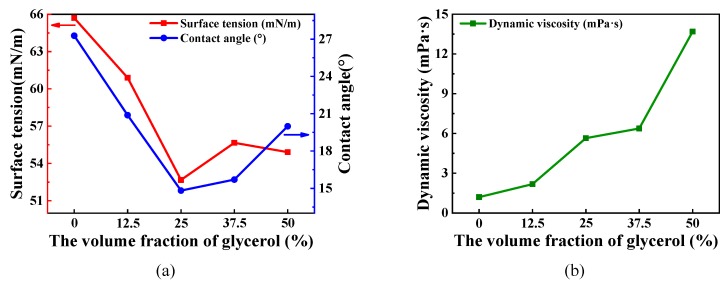
The surface tension and the contact angle and the dynamic viscosity of the all sample precursor solutions. (**a**) The surface tension and the contact angle. (**b**) The dynamic viscosity.

**Figure 2 micromachines-11-00311-f002:**
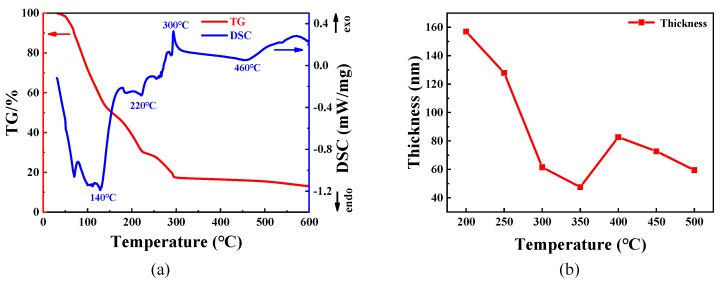
(**a**) The thermal gravimetric and differential scanning calorimetry (TG-DSC) curves of the sample solution with 25% volume fraction glycerol. The endothermic process and exothermic process are appointed as “endo” and “exo”, respectively. (**b**) The thickness of the WO_3_ films annealed at different temperatures.

**Figure 3 micromachines-11-00311-f003:**
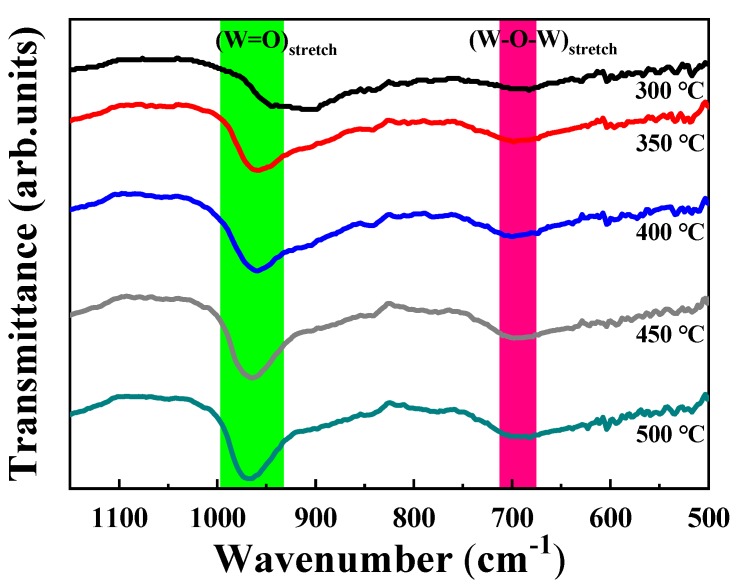
Fourier transform infrared (FT-IR) spectra of WO_3_ films prepared using the sample solution with 25% volume fraction glycerol.

**Figure 4 micromachines-11-00311-f004:**
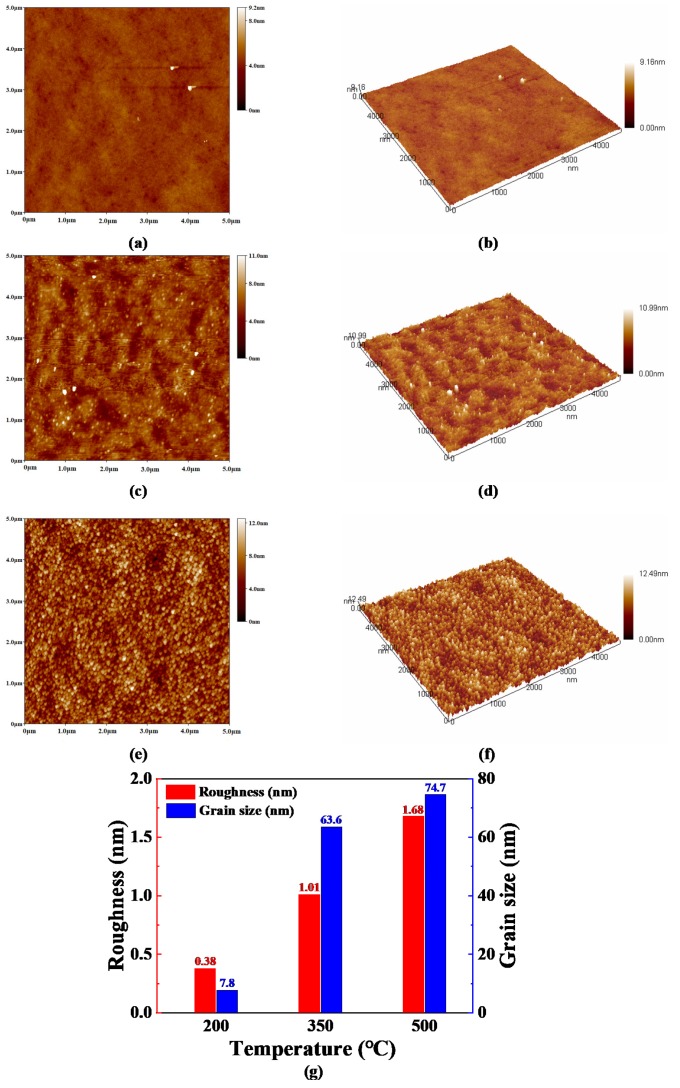
The atomic force microscope (AFM) 2D and 3D images of the WO_3_ thin films prepared using the sample solution with 25% volume fraction glycerol annealed at different temperatures: (**a**,**b**) 200 °C, (**c**,**d**) 350 °C, (**e**,**f**) 500 °C, respectively. (**g**) The surface roughness and the surface grain size of these films.

**Figure 5 micromachines-11-00311-f005:**
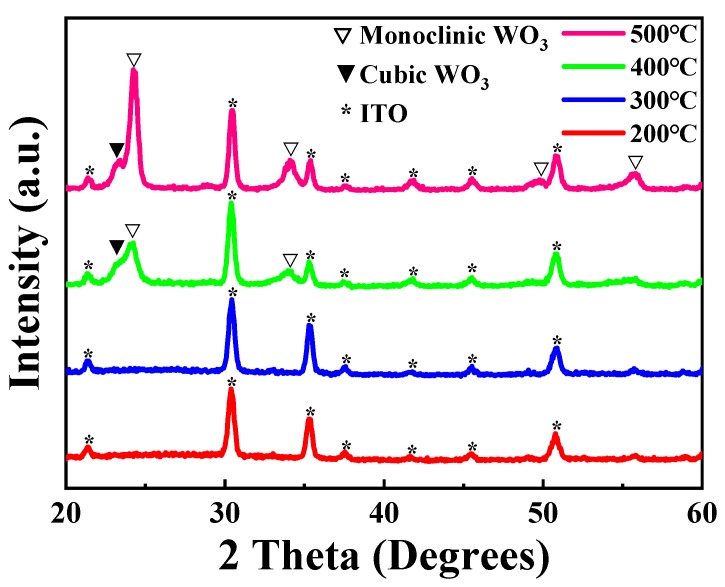
The X-ray Diffraction (XRD) patterns of the WO_3_ films prepared using the sample solution with 25% volume fraction glycerol annealed at different temperatures: 200 °C, 300 °C, 400 °C and 500 °C, respectively.

**Figure 6 micromachines-11-00311-f006:**
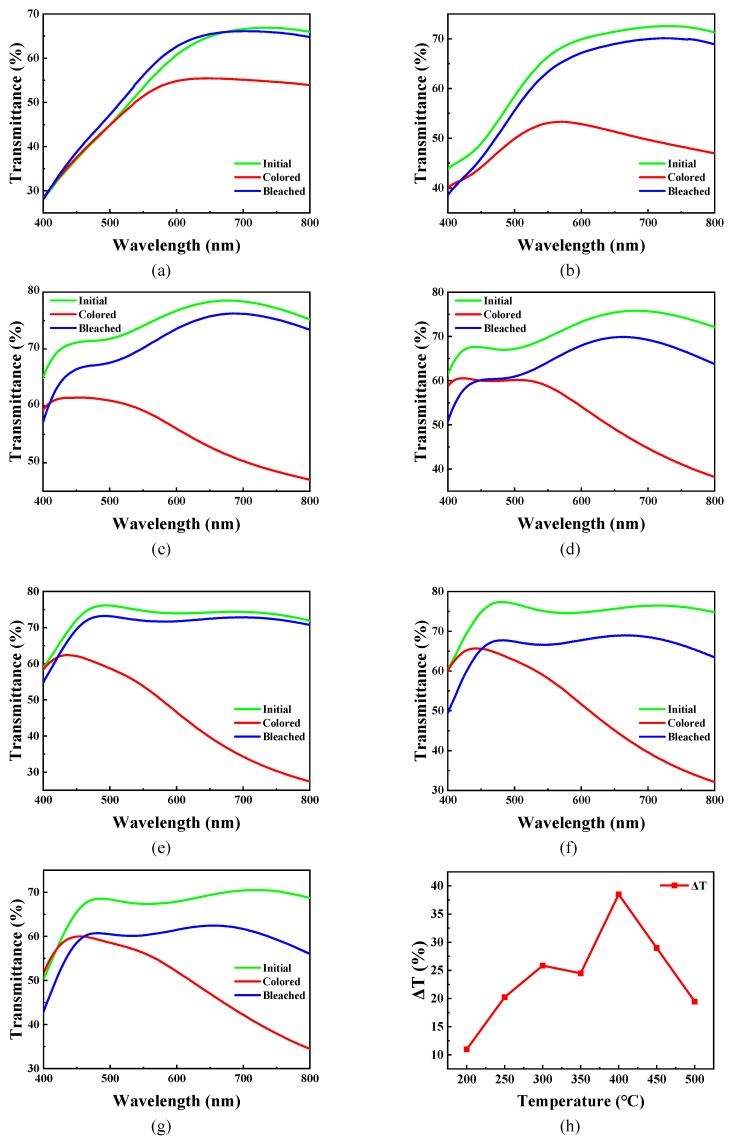
The optical transmittance spectra of the WO_3_ films annealed at different temperatures at the initial state, colored state and bleached state: (**a**) 200 °C, (**b**) 250 °C, (**c**) 300 °C, (**d**) 350 °C, (**e**) 400 °C, (**f**) 450 °C, (**g**) 500 °C, respectively. (**h**) The optical transmittance modulation ability (ΔT at 700 nm wavelength) with ±3.0 V voltages.

**Figure 7 micromachines-11-00311-f007:**
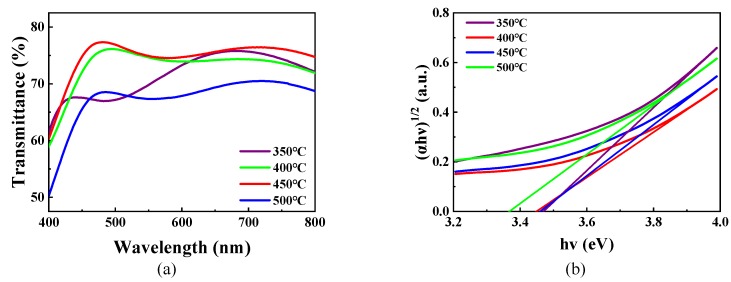
(**a**) The transmittance of WO_3_ films prepared using the sample solution with 25% volume fraction glycerol annealed at different temperatures. The transmittance of air is specified as 100%. (**b**) The curve of (αhν)^1/2^ versus hν for WO_3_ films.

**Figure 8 micromachines-11-00311-f008:**
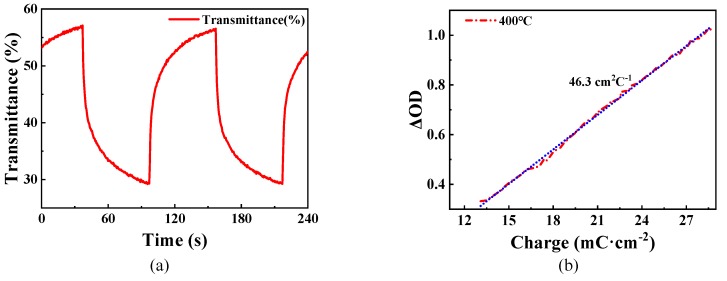
(**a**) Change curve of transmittance at 680 nm of WO_3_ films prepared using the sample solution with 25% volume fraction glycerol. (**b**) Variation of the optical density (OD) versus charge density for WO_3_ film. The applied voltage was range from −3.0 V to +3.0 V.

**Table 1 micromachines-11-00311-t001:** The comparison of optical band gap and coloration efficiency between this work and the reported works by synthesis methods.

Result from	Synthesis Method	Crystallinity	Optical Band Gap(eV)	Coloration Efficiency (cm^2^/C)
[5]	DC magnetron sputtering	Amorphous	3.51	40.5
[37]	Sol-gel	Amorphous	3.31	45.3
[37]	Sol-gel	Crystalline	3.10	27.4
[52]	Sol-gel	Crystalline	3.01	46.7
This work	Sol-gel free	Crystalline	3.37	46.3

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
