# Peer review of "Effect of the Ammonium Tungsten Precursor Solution with the Modification of Glycerol on Wide Band Gap WO3 Thin Film and Its Electrochromic Properties"

_micromachines, 2020, doi:10.3390/mi11030311_

Round 1
Reviewer 1 Report
The submitted work reports synthesis of WO3 films by inorganic and sol-gel free ammonium tungstate with the modification of glycerol using spin coating technique. The structure and optical properties of the as-synthesized films with different annealing temperatures are investigated. Moreover, the electrochromic performance of the films is reported. This work might be of potential interesting to readers worked in this filed. Several revisions are requested before it can be accepted for publication.
- The WO3 is an important oxide for various scientific device applications. It has diverse crystal structures and can be prepared through various methods to obtain a crystalline structure. Notably, the structure of the crystalline WO3 films in this study is in a mixed crystallographic structure (monoclinic and cubic phases). In the introduction section, I would suggest that the authors to highlight the importance of WO3 crystals with various crystal structures by different synthesis methods for diverse scientific device applications. The authors might be interested in including the following reference works in the introduction section.
-Nanomaterials , 10, 398 (2020).
-Nanomaterials 9, 669 (2019).
-Materials 11, 1627 (2018)
-Coatings 9, 90-100 (2019).
-CrystEngComm,17, 6070-6093 (2015).
- It would be better for the authors to include the average surface grain size from SEM images or AFM images in the revised version to understand the surface feature of the films.
- In general, 3D AFM images of thin films are not widely accepted in scientific journals. It would be better to show 2D AFM images of the films.
- Please use histogram to replace Fig. 4d. It would be more appropriate to show the tendency of surface roughness of the films. Or you need to show roughness values of all samples in the original Fig. 4d.
- What is the crystalline temperature for the WO3 films herein? Above 400C or between 300-400C (350C) ? The authors should clarify this information in the XRD section.
- Since the authors can measure contact angle, it is suggested that the contact angle of the solid WO3 films could be performed to understand the surface state of various annealed WO3 films herein.
- Line 213; Fig. 8 should be corrected to Fig. 7.
- The authors should pay attention to the fact that the WO3 films annealed at the elevated temperatures have a mixed crystal structure in this work. In general, WO3 has different bandgaps with different crystal structures. The phrases of bandgap section need to reword. I think the authors cannot separate the contribution from monoclinic and cubic phase from the current UV-Vis measurements for bandgap calculations. “The average bandgap of the WO3 films with a mixed crystal structure is …” would be an appropriate statement.
- In Fig. 6h, Why transmittance modulation ability slightly decreased at 350C ? What happened?
- How about the electrochromic reliability of the as-synthesized samples herein? The authors should comment on at least the optimal sample in this regard in the revised version.
Author Response
Dear reviewer,
We sincerely appreciate the valuable and professional suggestions by you for our manuscript entitled “Effect of the ammonium tungsten precursor solution with the modification of glycerol on wide band gap WO3 thin film and its electrochromic properties”. We have carefully addressed our manuscript based on the comments and suggestions, and the changes to the original manuscript are highlighted in the revised version by using the "Track Changes" function in Microsoft Word. Please check the revision.

Reviewer 2 Report
The manuscript (micromachines-729990) shows the effect of the ammonium tungsten precursor solution with the modification of glycerol on wide band gap WO3 thin film and its electrochromic properties. Authors present quite comprehensive analysis chemically. However, some comments still need to be addressed before further confirm the outcomes, shown as following:
- What is the WO3 thickness in the main analysis part (e.g. in Fig. 6)? Does the thickness affect the optical transmittance results? How authors can confirm that the thickness for each annealed samples (please provide the experimental measurement results to support this point, e.g. SEM images for thickness layer information check)? Authors should clarify this part in different anneal temperature effect.
- In Fig. 6, the figure caption should be edit and revised.
- This referee is curious on electrical part. What would be the resistance change between those annealed samples (200C-500C)? Are they the same or difference?
- Authors should provide a simple benchmark Table to compare the performance (coloration efficiency, optical band gap, etc...) with other fabricated methods and indicate what would be the main contribution in this work before jumping to conclusion.
Due to the above comments, this referee would like to put the manuscript status as "Major Revision" in current phase.
Author Response
Dear reviewer,
We sincerely express our thanks to you for the valuable and professional suggestions for our manuscript entitled “Effect of the ammonium tungsten precursor solution with the modification of glycerol on wide band gap WO3 thin film and its electrochromic properties”. We have carefully addressed our manuscript based on the comments and suggestions, and the changes to the original manuscript are highlighted in the revised version by using the "Track Changes" function in Microsoft Word. Please check the revision.

Round 2
Reviewer 1 Report
The authors have addressed all issues. It is acceptable for publication.
Author Response
Dear reviewer,
We sincerely appreciate the valuable and professional comments by you for our manuscript entitled “Effect of the ammonium tungsten precursor solution with the modification of glycerol on wide band gap WO3 thin film and its electrochromic properties”.
Reviewer 2 Report
This is the 3rd revision process. In the first question (2nd review), this referee would hope authors should add the thickness effect on the optical transmittance modulation ability in the manuscript. Although the results are partially affected, it would be nice to clarify the thickness indeed impact here, and mentioned the potential explaination in manuscript. And one more thing, how to separate the "impact factor" on thickness effect, binding structure effect, and crystallinity effect (all related to annealed samples) on optical transmittance modulation ability? Authors should address this point quite seriously to make it crystal clear for this referee and potential readers.
Author Response
Dear reviewer,
We sincerely appreciate the valuable and professional comments by you for our manuscript entitled “Effect of the ammonium tungsten precursor solution with the modification of glycerol on wide band gap WO3 thin film and its electrochromic properties”. We have carefully addressed our manuscript based on the reviewers’ comments, and the changes to the original manuscript are highlighted in the revised version by using the "Track Changes" function in Microsoft Word. Below please check for our response to the reviewers’ comments point-by-point, which we reflect in our revised manuscript.

Round 3
Reviewer 2 Report
Authors have replied this reviewer comments in detail. No additional comments from this referee.